# Gastrointestinal Helminths in Wild Felids in the Cerrado and Pantanal: Zoonotic Bioindicators in Important Brazilian Biomes

**DOI:** 10.3390/ani14111622

**Published:** 2024-05-30

**Authors:** Iago de Sá Moraes, Victória Luiza de Barros Silva, Beatriz Elise de Andrade-Silva, Ana Paula Nascimento Gomes, Nicoly Ferreira de Urzedo, Vitória Breda Abolis, Renata de Souza Gonçalves, Karina Varella Arpon, Zara Mariana de Assis-Silva, Lizandra Fernandes da Silva, Ellen Amanda Zago, Michelle Benevides Gonçalves, Ísis Assis Braga, Klaus Casaro Saturnino, Edson Moleta Colodel, Arnaldo Maldonado Júnior, Richard de Campos Pacheco, Dirceu Guilherme de Souza Ramos

**Affiliations:** 1Laboratório de Parasitologia e Análises Clínicas Veterinária, Instituto de Ciências Agrárias, Universidade Federal de Jataí, Jataí 75801-615, GO, Brazil; iago.samoraes@gmail.com (I.d.S.M.); nicknicoly2114@gmail.com (N.F.d.U.); viccabolis@gmail.com (V.B.A.); zaramariana@discente.ufj.edu.br (Z.M.d.A.-S.); lizandra.fsilva@hotmail.com (L.F.d.S.); isis@unifimes.edu.br (Í.A.B.); 2Laboratório de Parasitologia Veterinária e Doenças Parasitárias dos Animais Domésticos e Silvestres, Faculdade de Medicina Veterinária, Universidade Federal de Mato Grosso, Cuiabá 78060-900, MT, Brazil; victoria.luizabarros@hotmail.com (V.L.d.B.S.); ellen.amanda0605@gmail.com (E.A.Z.); michellebgoncalves@gmail.com (M.B.G.); richard@ufmt.br (R.d.C.P.); 3Laboratório de Biologia e Parasitologia de Mamíferos Reservatórios, Instituto Oswaldo Cruz, Rio de Janeiro 21040-360, RJ, Brazil; biaelisebio@gmail.com (B.E.d.A.-S.); apngomes@yahoo.com.br (A.P.N.G.); rnataps24@gmail.com (R.d.S.G.); karinavarella@gmail.com (K.V.A.); maldonad@ioc.fiocruz.br (A.M.J.); 4Laboratório de Anatomia Patológica Veterinária, Instituto de Ciências Agrárias, Universidade Federal de Jataí, Jataí 75801-615, GO, Brazil; klaus.sat@ufj.edu.br; 5Laboratório de Patologia Veterinária, Faculdade de Medicina Veterinária, Universidade Federal de Mato Grosso, Cuiabá 78060-900, MT, Brazil; edson.colodel@ufmt.br

**Keywords:** biodiversity, conservation, helminthology, parasitism, public health, zoonosis

## Abstract

**Simple Summary:**

Simple Summary: Wildlife in the Pantanal and Cerrado regions of Brazil face increasing threats, such as deforestation, urbanization, and road construction, which disrupt their natural habitats and increase the risk of diseases, including those that can spread to humans. The aim of this study was to characterize parasites affecting wild felids (large cats) in these areas, focusing on interactions among parasites, hosts, and the environment. The results provide a basis for the development of strategies to prevent the spread of disease and promote animal and human health. The use of advanced technology to monitor ecological changes and the importance of involving local communities in conservation efforts are emphasized. By integrating scientific research with public health measures and community engagement, this project aims to create sustainable solutions to protect biodiversity and public health. This is crucial for maintaining the ecosystem balance and ensuring the health of wildlife and nearby human populations.

**Abstract:**

Environmental changes in the Brazilian Pantanal and Cerrado facilitate the spread of parasitic diseases in wildlife, with significant implications for public health owing to their zoonotic potential. This study aimed to examine the occurrence and diversity of gastrointestinal parasites in wild felids within these regions to assess their ecological and health impacts. We collected and analyzed helminth-positive samples from 27 wild felids using specific taxonomic keys. Diverse parasitic taxa were detected, including zoonotic helminths, such as *Ancylostoma braziliense*, *Ancylostoma caninum*, *Ancylostoma pluridentatum*, *Toxocara cati*, *Toxocara canis*, *Dipylidium caninum*, *Taenia* spp., *Echinococcus* spp., and *Spirometra* spp. Other nematodes, such as *Physaloptera praeputialis* and *Physaloptera anomala,* were identified, along with acanthocephalans from the genus *Oncicola* and a trematode, *Neodiplostomum* spp. (potentially the first record of this parasite in wild felids in the Americas). Human encroachment into natural habitats has profound effects on wild populations, influencing parasitic infection rates and patterns. This study underscores the importance of continuous monitoring and research on parasitic infections as a means of safeguarding both wildlife and human populations and highlights the role of wild felids as bioindicators of environmental health.

## 1. Introduction

The Cerrado and Pantanal are critical biomes in Brazil, renowned for their vast biodiversity. The Pantanal, an extensive wetland ecosystem, is distinguished by its seasonal flooding, which fosters a mosaic of aquatic and terrestrial habitats. This dynamic environment supports substantial populations of species that are rare or endangered elsewhere in South America. In contrast, the Cerrado is a conservation hotspot, characterized by its remarkable species richness and high levels of endemism. This expansive tropical savanna biome exhibits a complex mosaic of vegetation types, including grasslands, shrublands, and forests. The Cerrado’s sparse terrestrial vegetation and relatively dry climate further contribute to its unique ecological characteristics [1,2,3,4,5].

Human activities, notably urbanization and agricultural expansion, have profound effects on these biomes, leading to habitat fragmentation, and have substantial ecological consequences [6]. Fragmentation reduces the biodiversity and population sizes of native felid species; it impedes their ability to locate prey, suitable territories, and mating partners, thus intensifying intraspecific competition and affecting survival [7,8].

Anthropogenic activities, including deforestation, have been particularly rapid and aggressive in flat biomes, such as the Cerrado and Pantanal, endangering the viability of keystone species, such as the jaguar (*Panthera onca*) and ocelot (*Leopardus pardalis*). These species play critical roles in maintaining an ecological balance; however, they face severe survival challenges owing to environmental alterations [9,10].

Furthermore, human-induced environmental changes increase the incidence of diseases, notably zoonotic infections, with approximately 61% of human pathogens being zoonotic and 71.8% of emerging human diseases originating from wildlife [11,12,13,14]. Concurrently, the process of spillback or zooanthroponosis complicates wildlife conservation efforts, presenting a novel concern about transmission of pathogens from humans to wild animals [15].

In particular, urban, peri-urban and rural development in wild areas increase the risk of pathogen transmission between wildlife, domestic animals and humans. With the increasing interactions among humans, domestic animals, and wildlife, a greater “spillover” of diseases from domesticated animals to wildlife is expected to occur [16]. Furthermore, when these diseases spill over into wildlife, wild animals can become reservoirs and amplifiers of these diseases, posing a threat back to domestic animals and humans [17,18]. The increased prevalence of parasitic diseases necessitates studies of parasite distributions, occurrence, and transmission dynamics [16,19,20]. Although much of our understanding of disease spillover focuses on bacteria and viruses, larger parasites such as helminths are also important but have received less attention [21].

This study aimed to identify gastrointestinal helminths in wild felids of the Cerrado and Pantanal to enhance our understanding of the interactions between human activity and animal health, with a particular focus on parasites that have zoonotic potential. This study reveals that wild felids in the Brazilian Pantanal and Cerrado are hosts to a diverse range of gastrointestinal parasites, many of which have zoonotic potential and pose significant health risks to humans. Environmental changes driven by human activities, such as deforestation and urbanization, intensify these risks by altering natural habitats and facilitating increased interactions among wildlife, humans, and domestic animals. These findings highlight the urgent need for integrated wildlife conservation and public health strategies that address the ecological impacts of human encroachment to safeguard both ecosystem health and human well-being.

## 2. Materials and Methods

This study was performed in the Cerrado and Pantanal biomes of Brazil. Both biomes are located in central-western Brazil, with the Cerrado extending into the states of Goiás, Mato Grosso, Mato Grosso do Sul, Minas Gerais, Bahia, Maranhão, Piauí, Rondônia, Paraná, and São Paulo e Distrito Federal, while the Pantanal is primarily situated in Mato Grosso and Mato Grosso do Sul. The study area included the Cerrado regions within the states of Mato Grosso and Goiás, as well as the Pantanal region within Mato Grosso (Figure 1).

Wild felids from these biomes were analyzed after being retrieved from wildfires, from roadkill incidents on the peri-urban highways of southwest Goiás, or after dying during care at the Veterinary Hospital of the Federal University of Mato Grosso (UFMT) in 2020–2024. Some animals cared for at the Veterinary Hospital were from the UFMT Biological and Research Institute, which is located within the university campus (Cuiabá, Mato Grosso). The carcasses were subjected to parasitological necropsy, during which all gastrointestinal contents were sifted using a 0.2 mm mesh Tamis-type stainless-steel sieve. The filtrates were immediately inspected under a stereoscopic microscope to collect parasites. Helminth specimens were preserved in 70% ethanol.

For morphological identification, the nematodes were hydrated and clarified in either 50% glycerol or lactophenol solution for up to 24 h. For more robust helminths, a 90% phenol solution was used, according to the methodology outlined by Hoffmann [22]. Acanthocephalans and cestodes were stained with carmine acid, followed by decolorization in 0.5% hydrochloric acid (HCl) in 70% ethanol, sequential dehydration in a graded alcohol series, and clarification using eugenol (4-allyl-2-methoxyphenol), according to a modified protocol described by Amato [23].

Temporary microscopic slides were prepared and evaluated using an optical microscope (Zeiss Microscope AXIO Scope A1, Carl Zeiss, Oberkochen, Germany) at magnifications ranging from 100× to 400×. The morphological structures of the helminths were documented using TCapture Imaging software version 5.1.1.0. Taxonomic classification of the parasites relied on specific taxonomic keys and descriptions [24,25,26,27,28,29,30,31,32,33,34,35].

This study was approved by the Ethics and Animal Use Experimentation Committee at UFJ (CEUA/UFJ) under protocol number 004/2022 and by the Ethics Committee on Animal Research of the Federal University of Mato Grosso (CEUA protocol no. 23108.015878/2019-65). Additionally, procedures in this study were previously approved by the “Instituto Chico Mendes de Conservação da Biodiversidade” (ICMBio permit no. 84201-2 and 55104-1). This ensures compliance with ethical guidelines and registration standards, facilitating research aimed at enhancing our understanding of the parasitological effects on these endangered felid populations.

## 3. Results

### 3.1. Hosts and Municipalities

The animals included in this study represent a significant diversity of feline species inhabiting the Pantanal and Cerrado biomes. *Leopardus pardalis* (ocelot), *Panthera onca* (jaguar), *Puma concolor* (cougar), and *Herpailurus yagouaroundi* (jaguarundi) are native species to these biomes, each with distinct ecological habits and niches, providing a representative sample for the study of parasitic infestations in these environments. In total, four host species were studied. Brazil hosts 10 wild felid species, but two of these species do not occur in the Cerrado and Pantanal biomes.

Infestations were detected in 27 wild felids, including *L. pardalis* (8), *P. onca* (8), *P. concolor* (5), and *H. yagouaroundi* (6), collected from peri-urban areas in the municipalities of Poconé (MT) in the Pantanal and Tangará da Serra (MT), Chapada dos Guimarães (MT), Cuiabá (MT), Jataí (GO), and Mineiros (GO) in the Cerrado biome (Figure 2).

### 3.2. Parasite Identification and Load

Comprehensive collection and taxonomic identification of 896 gastrointestinal helminth specimens from these felids revealed 14 species across 10 genera. The highest parasitic load was observed for *Ancylostoma pluridentatum* (n = 363) in *H. yagouaroundi*, followed by *Toxocara cati* in *L. pardalis*. In contrast, the lowest parasitic loads, with only one recovered specimen, were associated with *Ancylostoma braziliense* and *Spirometra* spp. in *P. concolor*, *Dipylidium caninum* and *Oncicola* spp. in *H. yagouaroundi*, and *Spirometra* spp. in *P. onca* (Figure 3). An analysis of the parasitic load distribution by biome indicated that parasite recovery was significantly higher in the Cerrado (n = 732 parasite by 21 hosts) than in the Pantanal (n = 164 by 6 hosts), and species diversity was higher in the Cerrado (n = 13 by 21 hosts) than in the Pantanal (n = 6 by 6 hosts; Figure 4).

### 3.3. Taxonomic Distribution

The genus *Oncicola* (Acanthocephala) was identified across all host species and in both studied biomes, with the highest parasitic loads in *P. onca* (n = 48) within Cerrado, and *L. pardalis* (n = 32) within the Pantanal. *L. pardalis* showed the highest frequency of parasitism by this genus (n = 4; Table 1).

Nematodes, such as *Ancylostoma*, *Toxocara*, *Physaloptera*, and *Trichuris*, were also identified. Ancylostomatidae was found in one *H. yagouaroundi* and two *P. concolor* from the Cerrado, including *A. braziliense* and a co-infection involving *A. pluridentatum* and *Ancylostoma caninum* in *P. concolor*. *Toxocara* was the most frequently encountered helminth genus, with *Toxocara cati* or *Toxocara canis* present in all host species and biomes. *T. cati* was particularly pervasive and infected 12 hosts, whereas *T. canis* was observed exclusively in two *P. onca*. *Physaloptera anomala,* and *Physaloptera praeputialis* were detected on *L. pardalis* and *H. yagouaroundi*/*P. onca*, respectively. *Trichuris vulpis* was isolated from a single *L. pardalis* in the Cerrado (Table 1).

Cestodes were identified across three Cyclophyllidea genera (*Echinococcus*, *Taenia*, and *Dipylidium*) and one Pseudophyllidea genus (*Spirometra*), with occurrence noted in both biomes and across all studied host species. *Spirometra* spp. and *Taenia* spp. were present in four of the nine cestode-parasitized hosts, including a co-infection in *P. concolor* from the Cerrado. However, *D. caninum* and *Echinococcus* spp. were found only in one host from the Cerrado (Table 1). The only identified trematode genus, *Neodiplostomum*, was found in one *L. pardalis* sample from the Cerrado. The parasitic diversity across host species was uniform in *L. pardalis*, *P. onca*, and *P. concolor*, each hosting seven different parasite species, whereas *H. yagouaroundi* had six parasite species (Table 1).

### 3.4. Co-Infection Patterns

An analysis of the complex patterns of co-infection among wild felids revealed 10 unique co-infection schemas. Within the Cerrado biome, *L. pardalis* was susceptible to multiple parasitic combinations, including co-infections with *T. cati*, *T. vulpis*, and *Neodiplostomum* spp. as well as other associations, such as *Oncicola* spp., *Taenia* spp., and combinations of *T. cati* and *Oncicola* spp. Conversely, in the Pantanal biome, consistent co-infection patterns were observed in the same felid species, particularly *Spirometra* spp. and *Oncicola* spp. along with *P. anomala* and *Spirometra* spp. Different patterns of co-infections were also noted across other host species (Figure 5; Table 1).

*Panthera onca* exhibited two co-infections involving *T. cati* and *T. canis* in both biomes and one co-infection of *T. cati* and *P. praeputialis* in the Cerrado. Meanwhile, *P. concolor* showed co-infections of *A. braziliense* and *T. cati*; *A. pluridentatum* and *A. caninum*; and *Taenia* spp., *Spirometra* spp., and *T. cati* in the Cerrado. In contrast, no co-infections were observed in *H. yagouaroundi* (Figure 5; Table 1).

### 3.5. Morphological Insights

#### 3.5.1. Acanthocephala

*Oncicola* spp. were characterized by a globular, cylindrical anterior trunk that elongates towards the rear and a retractable proboscis equipped with six rows containing six hooks each, devoid of a cervical collar, a feature that clearly separates them from the genus *Prosthenorchis* (Figure 6A). The proboscis of *Oncicola* spp. was anchored within a single-walled receptacle through a short neck. The lemnisci, which were long and tubular, extended towards the posterior trunk segment and were occasionally coiled (Figure 6B). In males, the anatomy included a copulatory bursa, two testicles (one anterior and one posterior), and cement glands (Figure 6A,B). Females were differentiated by a well-defined uterus that extended into a bell-shaped structure that progressed to the vagina and ultimately to the vulva.

#### 3.5.2. Nematoda

In the genus *Toxocara*, the anterior segment was characterized by cervical alae, three prominent lips (Figure 7A–C), and as characteristic of the genera, it presents ventriculus that intercalated between the esophagus and the intestine, which is not present in the genus *Toxascaris* (Figure 7E,F). Sexual dimorphism was evident; males displayed externalized spicules that curved at the tail end and featured a digitiform process, whereas females possessed a tapered tail that lacked both curvature and a digitiform process (Figure 7D).

Distinct morphological features among *Toxocara* spp. were primarily evident in the structure of the cervical alae. *Toxocara canis* exhibited narrower alae with gradual termination (Figure 7B), whereas *T. cati* had broader alae with abrupt, arrowhead-like terminations (Figure 7A). Shared traits were also observed, including digitiform processes in both species (Figure 7D).

Ancylostomatids were characterized by well-defined buccal capsules and a muscular esophagus (Figure 8A–F). Sexual dimorphism within this genus was clearly observable; males exhibited a copulatory bursa and paired posterior spicules (Figure 8G,H), whereas females displayed a well-defined uterus, which may or may not contain eggs, and a tapered posterior end (Figure 8I). Differentiation among species within this genus is based on the placement, shape, and number of teeth within the buccal capsule as well as the characteristics of the dorsal rays in the male copulatory bursa.

*Ancylostoma pluridentatum* was distinguished by the two pairs of ventral teeth that emerge from either side of the ventral dental plate. In the mature specimens, only the internal teeth were prominent at the opening of the buccal capsule. Moreover, this species featured three pairs of hook-shaped projections along the dorsal edge of the oral opening, distinguishing *A. pluridentatum* from other species within the genus (Figure 8A,D).

*Ancylostoma caninum* was characterized by a deep buccal capsule adorned with three pairs of teeth on each side of the ventral margin, complemented by a pair of triangular dorsal teeth. This morphological trait is common among ancylostomatids, including *Ancylostoma buckleyi*. However, *A. caninum* was uniquely identified by the presence of centrolateral teeth, which distinguished the species from its congeners (Figure 8B,E). Conversely, *A. braziliense* exhibited two dental plates, each bearing a single tooth (Figure 8C,F). A comparative analysis of the copulatory bursae revealed subtle variation among species, primarily in the length of the dorsal rays (Figure 8D,E).

The genus *Physaloptera* was distinguished by two large triangular lateral lips equipped with teeth on the anterior extremity, and the anterior cuticle displayed a cephalic collar (Figure 9A,B). Sexual dimorphism within this genus was marked by the presence of caudal alae and sessile papillae in males (Figure 9C,D), whereas females exhibited well-defined uteri, which were prominently visible when containing eggs (Figure 9E,F).

We detected *P. praeputialis* and *P. anomala* in *H. yagouaroundi*/*P. onca* and *L. pardalis*, respectively. *Physaloptera praeputialis* exhibited an anterior region featuring triangular lips with small teeth and a cuticular sheath that reflects forward at the anterior end, forming a preputial-like collar. The cuticle in both sexes extended posteriorly, forming a protective sheath that projects beyond the caudal terminus of the body (Figure 9G,H). This morphological configuration supports distinct ecological adaptations, facilitating survival and propagation of the parasite within its host.

*Physaloptera anomala* was characterized by the presence of three denticles on each lip (Figure 9A). Species differentiation was principally determined by the location and size of the sessile and pedunculate papillae in the posterior region of the male, together with the caudal bursa (Figure 9D). The arrangement of the pedunculated papillae included three pre-anal and one post-anal pair. Among the preanal papillae, the middle pair was positioned closest to the anus. The postanal region featured five pairs of papillae; the first and second pairs were small and aligned directly behind the anus, whereas the fourth and fifth pairs were larger and centrally located on the tail. The spatial distribution of these papillae was distinct; the gap between the third and fourth pairs was approximately four times that between the second and third pairs and double that between the fourth and fifth pairs.

*Trichuris vulpis* exhibited a marked morphological disparity between its anterior and posterior segments, with the anterior part being significantly thinner and more elongated, resembling a whip. The cuticle of this region was marked by transverse striae with a longitudinal bacillary stripe along the ventral esophageal region. The posterior segment in males featured a spiraled configuration and was equipped with an evaginated spiny sheath (Figure 10). In females, the tail end was subtly curved, and gravid individuals displayed uteri laden with brownish eggs, each capped with an operculum at both ends.

#### 3.5.3. Cestoda

*Spirometra* spp. exhibited a dorsoventrally flattened scolex with attachment bothria and notably lacked hooks. The proglottids were organized such that the vagina is centrally located, and the uterus is uniform and spirally shaped, typically filled with eggs (Figure 11A).

*Dipylidium caninum* was distinguished by its small scolex bearing four suckers and a rostellum equipped with multiple rows of hooks, enabling it to anchor firmly to the host’s intestinal mucosa. The proglottids of *D. caninum* were broader than they were long, and these dimensions were reversed during gravidity, adopting a barrel-like shape. Gravid proglottids were characterized by their segmentation into ovigerous sacs filled with eggs (Figure 11C), providing a clear indication of their reproductive status.

Species within the genus *Taenia* were distinguished by well-defined suckers on the scolex, and the examined specimens exhibited a rostellum equipped with hooks (Figure 11B,D). The body segments, or proglottids, were elongated and longer than they were wide, and each segment featured a hermaphroditic reproductive system that is characteristic of the genus. In contrast, *Echinococcus* spp. exhibited four suckers and a rostellum with hooks but included only three proglottids (Figure 11E), indicating a more streamlined morphological structure.

#### 3.5.4. Trematoda

*Neodiplostomum* (syn. *Fibricola*) spp. displayed a dorsoventrally flattened body with a distinct morphological division; the posterior part was fusiform, and the anterior part was spatulate (Figure 12A–D). The cuticle was adorned with fine spines (Figure 12B) that tapered towards the posterior end. This species had two suckers: an oral sucker at the front of the body, used for attachment to the host, and a ventral sucker positioned further back. The distribution of vitelline follicles was limited and did not extend beyond the area between the ventral sucker and anterior testis (Figure 12B–D). The digestive system began with a small pre-pharynx, leading to a globular pharynx, which then split into a bifurcated esophagus, illustrating the specialized feeding structure of this trematode.

## 4. Discussion

### 4.1. Parasite Occurrence and Consequences

Anthropogenic influences typically enhance the survival and proliferation of generalist helminths with direct life cycles, as these parasites are less dependent on a complex host system for life cycle completion than are specific helminths with heteroxenous cycles, which may be due to host scarcity [36]. This dynamic can be observed in our study, in which helminths from the genera *Ancylostoma* and *Toxocara* showed high frequencies.

Ancylostomatids, which are geohelminths with a direct lifecycle, are known to cause hemorrhagic gastroenteritis in carnivores. The primary mode of transmission is the oral–fecal route, which not only facilitates the spread of infection between hosts but also supports the occurrence of spillover and spillback events [37]. The detection of species such as *A. braziliense* and *A. caninum* in wild felids can possibly underscores their close interaction with human-altered landscapes that facilitates contacts of domestic and wild animals, highlighting the impact of anthropogenic activities on ecological health and parasite transmission dynamics [38,39,40].

These helminths are frequently found in both domestic and wild felids, indicating their broad zoonotic potential. Moreira et al. [41] documented the presence of Ancylostomatidae eggs in *P. onca* and *L. pardalis* in the Cerrado region of Brazil, underscoring the vulnerability of these wild species to parasites that are typically associated with domestic animals. Similar findings have been reported for other wild felids, such as *P. concolor* and *Leopardus wiedii* (margay), further illustrating the widespread nature of these infections [42,43].

The transmission of these parasites is often facilitated by environmental contamination through the dispersion of eggs from infected hosts [44]. For *Toxocara* spp., specifically *T. canis* and *T. cati*, which commonly infect wild carnivores, vertical transmission from the mother to offspring during gestation and via maternal milk plays a crucial role in maintaining the high prevalence and dissemination of these helminths [45,46]. *Toxocara canis* infection observed in felids may be correlated with the fact that these animals were from the UFMT Biological and Research Institute, where numerous dogs reside. This close interaction likely contributes to the occurrence and transmission of these helminths.

These infections have significant clinical implications. *A. braziliense* is frequently associated with dermatological conditions, whereas *A. caninum* is linked to eosinophilic enteritis and is a potential cause of diffuse unilateral subacute neuroretinitis in humans. These associations underline the close interrelationship between humans and domestic carnivores [47]. Infection with *Toxocara* spp. in humans can lead to granulomatous lesions in the eyes and neurological complications, including meningitis, encephalitis, myelitis, and cerebral vasculitis, as well as cutaneous and gastrointestinal issues [48,49,50]. Anthropization facilitates environmental interactions among dogs, cats, wildlife, and humans, thereby increasing the likelihood of cross-species transmission [51].

*Physaloptera*, a genus of parasitic nematodes within the order Spirurida and the family Physalopteridae, exhibits a broad geographic distribution and diverse hosts, including anteaters, jaguars, and pumas [52,53,54,55,56,57,58,59,60,61,62]. In Brazil, this parasite has been identified in native species, such as *Cerdocyon thous* (crab-eating fox) and *Chrysocyon brachyurus* (manned wolf) [43]. The life cycle of *Physaloptera* includes various arthropods as intermediate hosts, such as beetles, cockroaches, grasshoppers, and crickets [63,64,65], whereas reptiles, rodents, amphibians, and birds serve as paratenic hosts [66,67,68,69]. Wild carnivores may ingest these arthropods during periods of food scarcity or in response to hunting instincts triggered by arthropod movement [44,70]. Recent studies, such as Mendoza and Ortranto [71], suggest that *Physaloptera* infection in felids could represent a significant zoonotic risk, bridging the gap between wild and domestic species and potentially affecting humans through shared vectors or intermediate hosts. This highlights the need for ongoing research on parasite ecology and the implementation of preventive measures to curtail the risk of emerging zoonoses, particularly in areas where human habitation encroaches on natural habitats.

While *T. vulpis* is generally not considered a major zoonotic threat, its presence has been linked to visceral larva migrans syndrome and intestinal infections in humans [72,73,74,75]. Given that *T. vulpis* predominantly infects domestic animals, its detection in wild felids underscores the impact of human activities and the proximity of wild animals to peri-urban environments. This proximity leads to alterations in predator–prey dynamics, driving wild animals to encroach upon human settlements in search of food, thereby increasing exposure to parasites typically associated with domestic species [76].

*Dipylidium caninum*, a cestode affecting both domestic animals and humans, requires an arthropod in its life cycle. Infection in definitive hosts occurs through the ingestion of infected fleas, predominantly in the genera *Ctenocephalides* and *Pulex*, and lice in the genus *Thricodectes* [76,77]. Gravid proglottids of *D. caninum*, often found in the feces of hosts, are visually akin to rice grains during the shedding process [78]. Factors influencing the prevalence of *D. caninum* in domestic and wild carnivores include the host age and behaviors, such as shelter sharing, which can increase the susceptibility and severity of parasitism. In contrast, in wild carnivores, the prevalence of infection tends to decrease during periods of prey scarcity, particularly within migratory ecosystems, where fluctuations in prey availability are pronounced [79].

Research on the prevalence of gastrointestinal helminth infections in wild felids has demonstrated variation in infection rates. For instance, a study in London identified *D. caninum* as the second most prevalent helminth among 93 examined wild felids with a prevalence rate of 32.8% [80]. In contrast, a study in Australia reported a prevalence of only 2% in similar hosts [81]. Such variation may be attributed to the proximity of these animal habitats to urban environments, echoing the high prevalence rates among stray cats reported in studies of urban areas [82].

The genus Taenia presents significant zoonotic concerns. In wild felids, species such as Taenia taeniaeformis, Taenia pisiformis, Taenia omissa, Taenia macrocystis, and Taenia crassipoda have been documented in *P. concolor*, *Leopardus geofroyi* (Geofroy’s cat), and *L. wiedii*, among others [43,83,84,85]. Infection with these parasites can lead to cysticercosis in humans, highlighting the need for monitoring and preventive measures [8,86].

Within the genus Echinococcus, multiple species, including *Echinococcus canadensis*, *Echinococcus felidis*, *Echinococcus multilocularis*, *Echinococcus oligarthrus*, *Echinococcus granulosus*, *Echinococcus shiquicus*, and *Echinococcus vogeli*, have been identified in wild carnivores and possess significant zoonotic potential [87]. Notably, *E. oligarthrus* is prevalent among wild felids in South America, highlighting regional epidemiological trends that require ongoing monitoring and control efforts [43,88]. However, the morphological identification of *E. oligarthrus* and other species within genera Echinococcus is not precise, relying heavily on morphometric analysis, which itself can lack accuracy [86].

Although the life cycles and modes of transmission vary among species within the families Taeniidae and Diphyllobothriidae, they predominantly involve an indirect cycle linked to the ingestion of larval forms located in the musculature and subcutaneous tissues of intermediate hosts. This aspect underscores the integral role of carnivorous and hunting behaviors in facilitating parasitic infections in wild populations [86]. Several species within these groups have been implicated in parasitic zoonoses that pose significant health risks to humans and domestic animals [89].

Species in the genus Oncicola, including Oncicola campanulata, Oncicola chibigouzouensis, Oncicola oncicola, Oncicola lamacrurae, Oncicola venezuelensis, Oncicola magalhaesi, and Oncicola paracampanulata, are frequently found in wild felids [43,90,91,92]. A common challenge in the accurate identification of these species is the loss of characteristic hooks during parasite removal from the insertion tissue.

*Oncicola* spp. induce lesions within the small intestines of wild felids, potentially leading to substantial nutritional deficits. Under severe parasitism, coupled with food scarcity or weakened conditions, these infections may progress to serious disease or mortality [93,94]. Although the histopathological response to intestinal epithelial invasion by *Oncicola* spp. often involves minimal signs of inflammation, the frequent observation of collagenous tissue at the insertion sites indicates chronic infection stages [92].

The embedding of these parasites deep within the intestinal walls, penetrating the muscular layer, facilitates direct nutrient absorption through the body wall and lacunar channels in the hypodermis. The ecological and public health significance of *Oncicola* infections in wild felids is substantial and serves as an indicator of overall ecosystem health and biodiversity. Although these parasites are not directly transmissible to humans, their prevalence and severity in wild felids reflects broader environmental health dynamics [95].

*Neodiplostomum*, a genus associated with aquatic hosts, such as fish, birds, and certain small mammals, such as *Hydromys chrysogaster* (water rat or rakali), illustrates the impact of environmental changes, including deforestation and urbanization, on parasite–host dynamics [96,97,98]. These alterations influence the water distribution and, subsequently, the distribution of parasites. The observed morphological characteristics of this parasite in wild felids aligned with those of *Neodiplostomum* (syn. *Fibricola*) *minor* described by Dubois [35]. Molecular analyses are essential to confirm these findings; however, this observation may represent the first documented occurrence of this trematode in felids in the Americas.

Another species of Diplostomatidae, such as *Alaria* spp., has been described in felids of South America. However, the morphological traits of our helminth are not compatible with these species. Specifically, the forebody is no longer than the hindbody, there are no auricular pseudosuckers, the ovary is not located at the junction of the fore- and hindbody, and the vitellarium is not mainly confined to the forebody. Conversely, our helminth more closely resembles *Neodiplostomum* (syn. *Fibricola*) spp., which display a spatulate forebody and vitellarium usually located in the forebody but occasionally penetrating the hindbody [35].

### 4.2. Impacts of Environmental Changes and Co-Infections Considerations

In Brazil, wild felids are increasingly threatened by rapid environmental changes, including deforestation, urbanization, and road construction. These anthropogenic activities significantly alter ecological niches and reduce resource availability, affecting habitat stability and integrity [99,100,101]. As apex predators, felids play a fundamental role in controlling populations of their natural prey, thereby influencing the dynamics of the entire ecosystem [102]. Human-induced modifications to landscapes not only shrink natural habitats but also reduce the buffer zones between urban, rural, and wild areas, increasing the risk of interspecific parasite transmission, including those with zoonotic potential, and thereby altering parasite–host–environment dynamics [103,104].

Furthermore, the intersection of wild and human habitats, particularly in peri-urban zones, markedly increases the risk of zoonotic parasite transmission. This proximity enhances interactions between humans, domestic animals, and wildlife, consequently raising the potential for diseases such as echinococcosis (*Echinococcus* spp.), toxocariasis (*Toxocara* spp.), and other helminth infections [105,106]. Urban expansion further alters local ecological dynamics, influencing the distributions of hosts and vectors and reshaping patterns of parasitism [107].

Human activities may disrupt the ecological balance by facilitating the spread of vectors, intermediate hosts, and paratenic hosts of significant zoonotic parasites. This often results in higher concentrations of animal populations in confined areas, increasing the likelihood of completing the transmission cycles of these parasites [36].

Zhu et al. [108] investigated the influence of human population density and temperature variation on the prevalence of *Toxoplasma gondii* oocyst shedding by domestic and wild felids, illustrating how anthropogenic changes affect disease dynamics in these animal populations. This study highlights the complex interplay between environmental and anthropogenic factors that shape the epidemiological landscape of parasitic infections. Therefore, monitoring these parasites is critical to understanding ecological health and preventing zoonotic diseases. Comprehensive wildlife conservation and management strategies that prioritize ecosystem health are needed to mitigate the impacts of human encroachment and maintain biodiversity integrity.

The two biomes evaluated in this study, the Pantanal and the Cerrado, exhibit distinct ecological characteristics that support diverse parasitic communities. Environmental factors such as temperature, humidity, and seasonal variation significantly affect the prevalence and lifecycle dynamics of parasites [100]. In the Pantanal, periodic floods and drought distinctly modulate the occurrence of parasites, resulting in defined seasonal infection trends for species such as *Spirometra* spp. [100,106,109,110,111,112]. In contrast, the characteristics of the Cerrado influence the lifecycle of parasites that rely on direct contact with the soil or ingestion of infected prey, including *Taenia* spp. and *Ancylostoma* spp. [113,114,115].

The behaviors of wild felids, including their dietary habits and hunting ranges, determine their exposure to parasitic agents. Apex predators such as *P. onca* encounter a diverse array of parasites due to their varied diet, whereas smaller species such as *H. yagouaroundi* experience different parasitic risks due to their more restricted dietary habits, thus affecting the parasitic diversity observed within these hosts [13]. Immunological resilience and genetic diversity within feline populations are pivotal in determining their susceptibility or resistance to parasitic infections, potentially explaining the variability in the parasitic burden observed among hosts [116].

Co-infection with multiple parasitic species can significantly influence disease severity. The overall health impact of these infections can vary widely. Some combinations of parasites can even provide protective effects against severe disease manifestations, making it challenging to accurately assess the multispecies host–pathogen ecosystem [117,118,119,120,121]. However, it is important to note that our study relies on the necropsy of deceased animals, which could impact the results due to the decomposition of helminths. This may lead to an underestimation of infections and co-infections.

## 5. Conclusions

In our study, zoonotic species such as *T. cati*, *T. canis*, *A. braziliense*, *A. caninum*, *D. caninum*, and *Echinococcus* spp. were identified. Understanding the dynamics of parasitism in the Brazilian Pantanal and Cerrado requires a detailed examination of how parasites, hosts, and environmental factors interact. This interaction increases the potential for cross-species transmission of zoonotic pathogens, highlighting the urgent need for comprehensive monitoring programs. These programs should integrate animal health, public safety, and environmental conservation to effectively address the challenges posed by zoonotic diseases in these evolving ecosystems.

## Figures and Tables

**Figure 1 animals-14-01622-f001:**
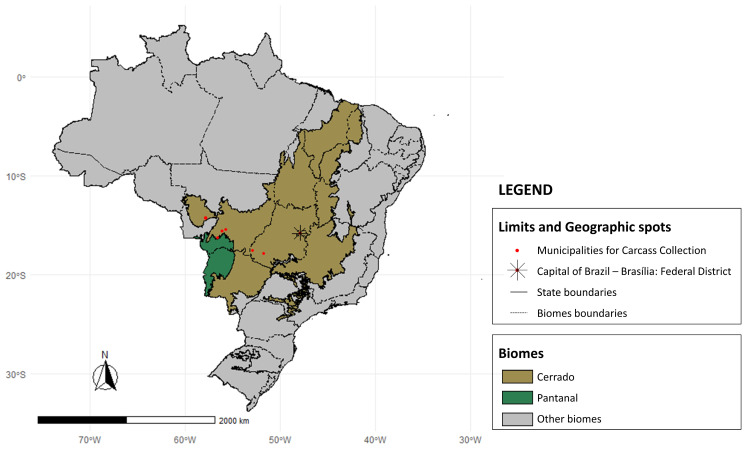
Geographical distribution of carcass collection of wild felids sites and biomes in Brazil.

**Figure 2 animals-14-01622-f002:**
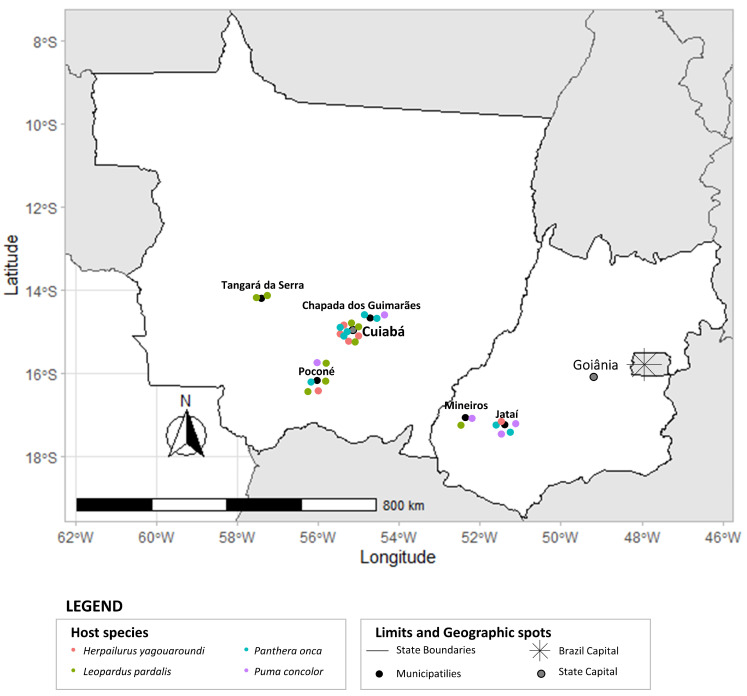
Municipalities in the Brazilian Cerrado and Pantanal biomes in which the remains of the wild felids were recovered for the analysis of gastrointestinal helminths are marked.

**Figure 3 animals-14-01622-f003:**
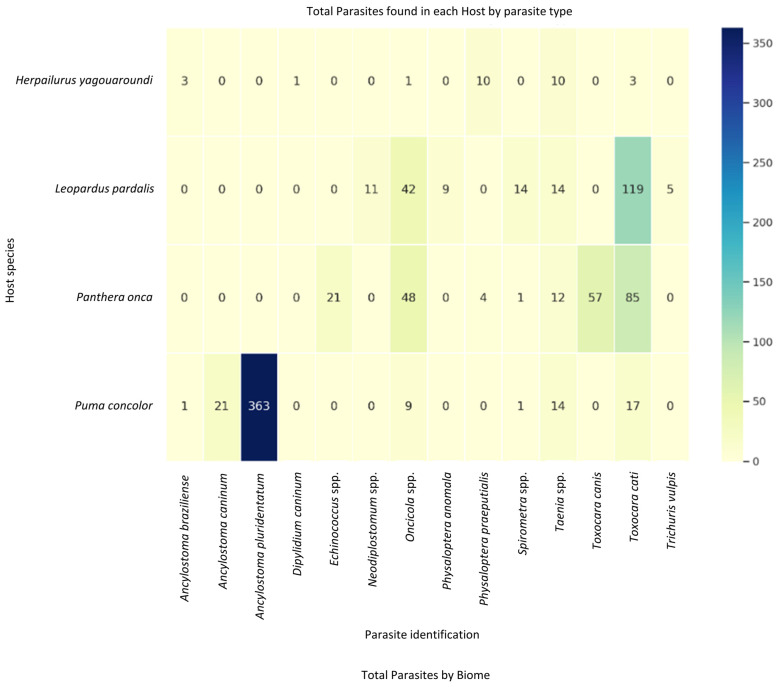
Heatmap of total parasites found in each host by parasite type and a bar plot of total parasites by biome. Light yellow indicates lowest range of parasite counts, indicating relatively minor infections, and dark blue represents the highest rang of parasite counts, indicating severe infections.

**Figure 4 animals-14-01622-f004:**
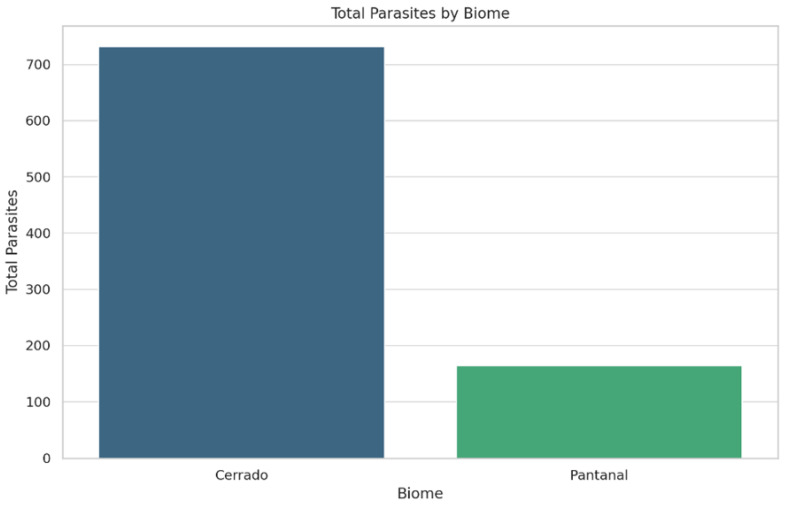
Analysis of parasite–host–environment relationships. Total of parasite count by Biome (Cerrado and Pantanal).

**Figure 5 animals-14-01622-f005:**
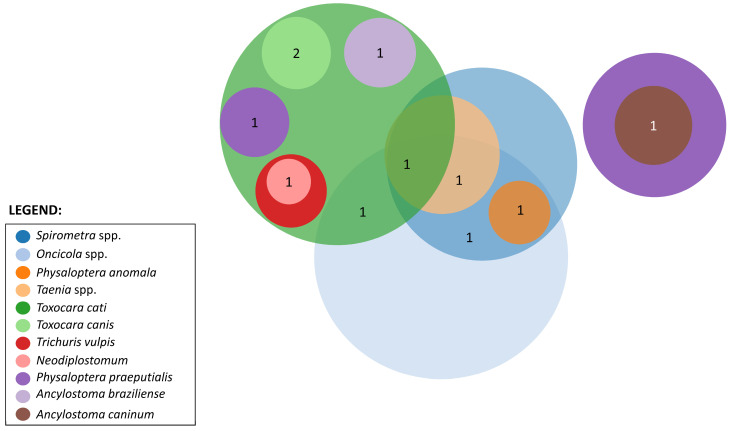
Venn diagram representation of helminth co-infections observed in the studied wild felids. Colors denote different helminth taxa, and the numbers indicate the frequency of observed co-infections among the studied wild felids.

**Figure 6 animals-14-01622-f006:**
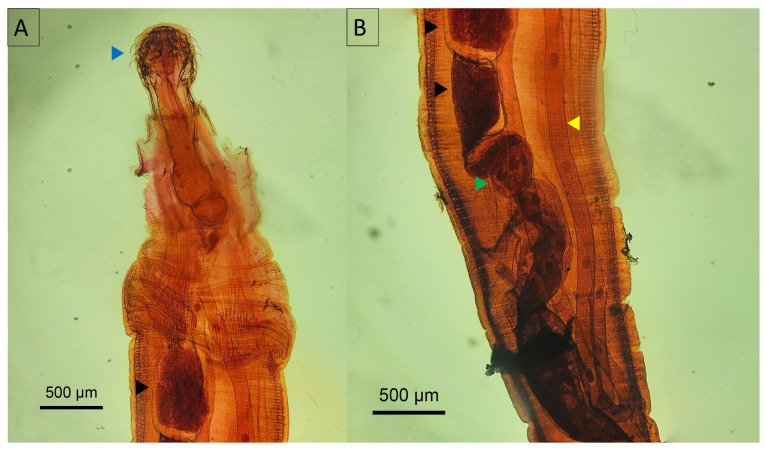
Micrograph of *Oncicola* spp. (**A**) Anterior portion of male *Oncicola* spp. recovered from Panthera onca; (**B**) posterior portion. Blue arrowhead—proboscis with hooks; black arrowhead—testis; yellow arrowhead—lemnisci; green arrowhead—cement glands.

**Figure 7 animals-14-01622-f007:**
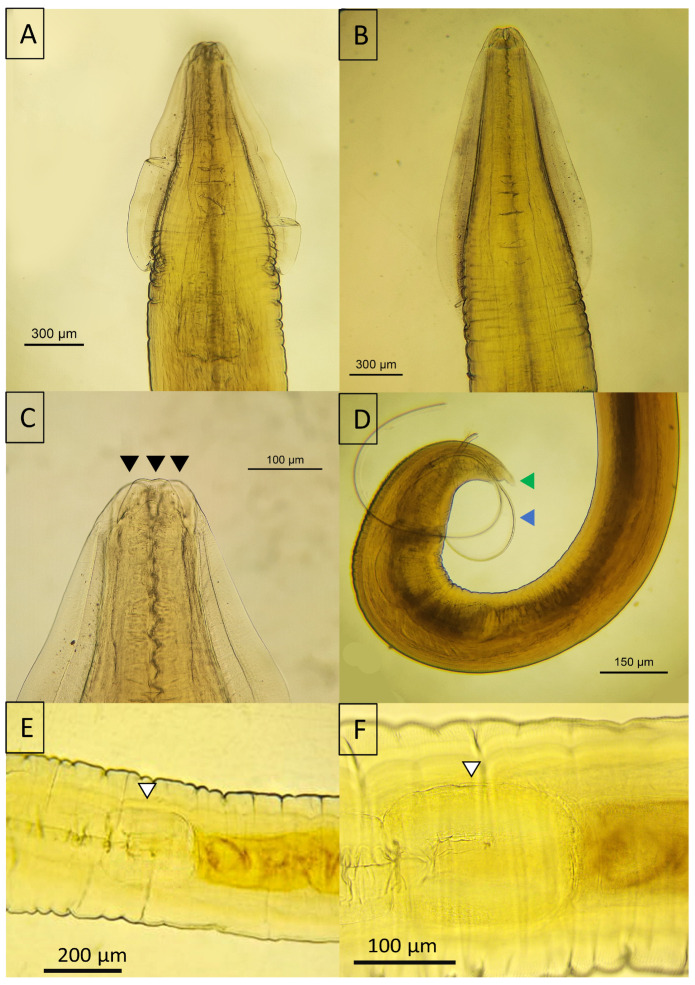
Micrographs of *Toxocara* spp. detected in *Panthera onca*. (**A**) Cervical wing of *Toxocara cati*; (**B**) cervical wing of *Toxocara canis*; (**C**) anterior view of *T. cati* showing the three lips; (**D**) posterior view of a male *T. canis* with both spicules externalized. (**E**,**F**) Ventriculus that intercalated between the esophagus and the intestine in *T. canis*. Black arrowhead—lips; blue arrowhead—spicules; green arrowhead—digitiform process; white arrowhead—ventriculus.

**Figure 8 animals-14-01622-f008:**
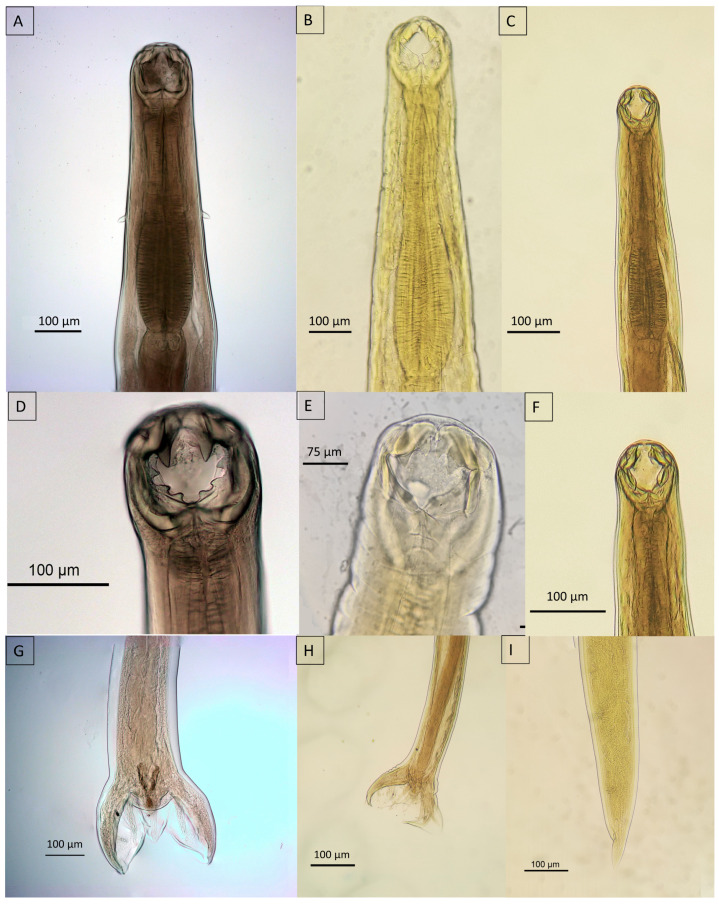
Micrograph of Ancilostomatids recovered from wild felids. (**A**,**D**) Anterior view of *Ancylostoma pluridentatum* from *Puma concolor*; (**B**,**E**) anterior view of *Ancylostoma caninum* from *P. concolor*; (**C**,**F**) anterior view of *Ancylostoma braziliense* from *Herpailurus yagouarandi*; (**G**) posterior view, male of *A. pluridentatum*; (**H**) posterior view, male of *A. braziliense*; (**I**) posterior view, female of *A. caninum*.

**Figure 9 animals-14-01622-f009:**
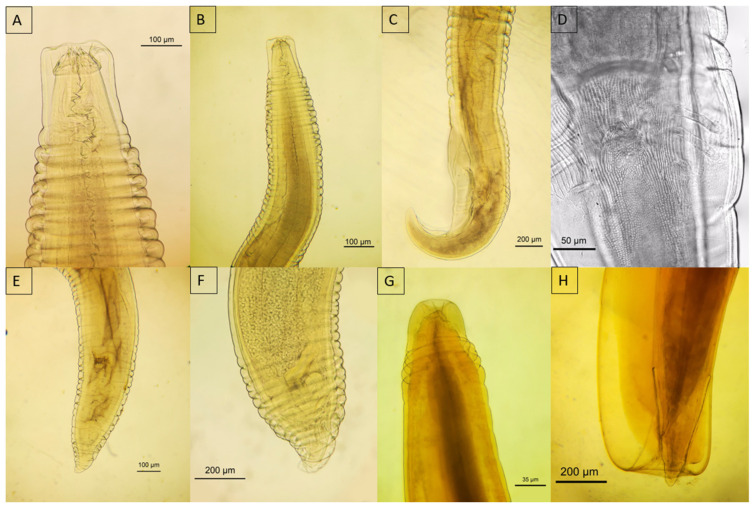
Micrograph of *Physaloptera* spp. recovered from wild felids. (**A**,**B**) Anterior portion of *Physaloptera anomala* from *Leopardus pardalis*; (**C**) posterior portion of male *P. anomala*; (**D**) showing the sessile papillae; (**E**) posterior portion of female *P. anomala*; (**F**) posterior end, part of the uterus filled with eggs; (**G**) anterior portion; (**H**) posterior portion of female showing the cuticular sheath of *Physaloptera praeputialis* in *Herpailurus yagouarandi*.

**Figure 10 animals-14-01622-f010:**
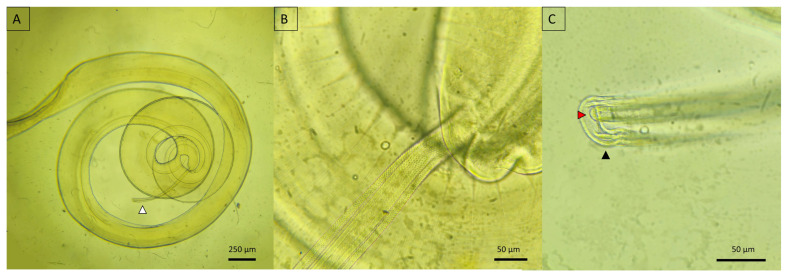
Micrograph of *Trichuris vulpis* recovered from wild felids. (**A**–**C**) Posterior portion of male *T. vulpis* from *Leopardus pardalis*; (**B**) highlighting the proximal portion of the sheath with spiny cuticle; (**C**) opening of the sheath with the tip of the spicule internalized. White arrowhead—spicule; black arrowhead—shealth; red arrowhead—tip of the spicule.

**Figure 11 animals-14-01622-f011:**
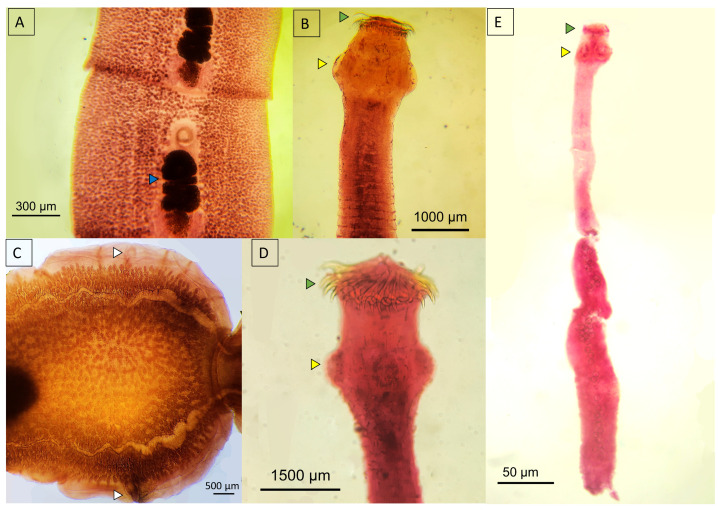
Micrograph of cestodes recovered from wild felids. (**A**) Mature proglottid of *Spirometra* sp. from *Leopardus pardalis*; (**B**,**D**) anterior portion of *Taenia* spp. from a *Herpailurus yagouarandi* and a *Panthera onca*, respectively; (**C**) gravid proglottid of *Dipylidium caninum* from *H. yagouarandi*; (**E**) *Echinococcus* sp. from *P. onca*. Blue arrowhead—uterus; yellow arrowhead—suckers; green arrowhead—hooks; white arrowhead—genital pore.

**Figure 12 animals-14-01622-f012:**
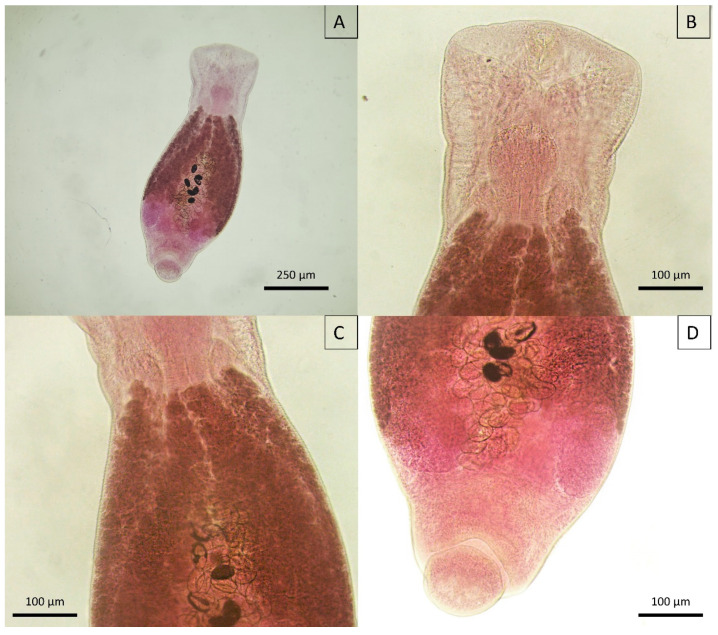
Micrograph of *Neodiplostomum* (syn. *Fibricola*) spp. recovered from *Leopardus pardalis*. (**A**) Adult *Neodiplostomum* spp.; (**B**) anterior portion; (**C**) middle portion with eggs and vitelline glands; (**D**) posterior portion highlighting the testes.

**Table 1 animals-14-01622-t001:** Occurrence of parasitism in wild felids by host species, biome, and geographic locality.

Host	Host ID	Biome	Location	Helminth Species	n(+)
*Herpailurus yagouaroundi*	UFJ-LPPV-156	Pantanal	Poconé, MT	*Physaloptera praeputialis*	10
	UFJ-LPPV-78	Cerrado	Cuiabá, MT	*Taenia* spp.	10
	UFJ-LPPV-80	Cerrado	Cuiabá, MT	*Dipylidium caninum*	1
	UFJ -LPPV-85	Cerrado	Cuiabá, MT	*Ancylostoma braziliense*	3
	UFJ-LPPV-86	Cerrado	Cuiabá, MT	*Oncicola* spp.	1
	UFJ-LPPV-304	Cerrado	Jataí, GO	*Toxocara cati*	3
*Leopardus pardalis*	UFJ-LPPV-133	Pantanal	Poconé, MT	*Spirometra* spp.	5
				*Oncicola* spp.	4
	UFJ-LPPV-154	Pantanal	Poconé, MT	*Toxocara cati*	5
	UFJ-LPPV-155	Pantanal	Poconé, MT	*Physaloptera anomala*	9
				*Spirometra* spp.	9
				*Oncicola* spp.	32
	UFJ-LPPV-216	Cerrado	Tangará da Serra, MT	*Oncicola* spp.	3
				*Taenia* spp.	14
	UFJ-LPPV-303	Cerrado	Cuiabá, MT	*Toxocara cati*	6
				*Trichuris vulpis*	5
				*Neodiplostomum* spp.	11
	UFJ-LPPV-302	Cerrado	Cuiabá, MT	*Toxocara cati*	75
	UFJ-LPPV-301	Cerrado	Cuiabá, MT	*Toxocara cati*	24
				*Oncicola* spp.	3
	UFJ-LPPV-75	Cerrado	Mineiros, GO	*Toxocara cati*	9
*Panthera onca*	UFJ-LPPV-140	Pantanal	Poconé, MT	*Toxocara cati*	62
				*Toxocara canis*	51
	UFJ -LPPV-77	Cerrado	Chapada dos Guimarães, MT	*Spirometra* spp.	1
	UFJ-LPPV-79	Cerrado	Cuiabá, MT	*Taenia* spp.	12
	UFJ-LPPV-82	Cerrado	Jataí, GO	*Toxocara cati*	10
	UFJ-LPPV-81	Cerrado	Jataí, GO	*Toxocara cati*	5
				*Physaloptera praeputialis*	4
	UFJ-LPPV-83	Cerrado	Cuiabá, MT	*Oncicola* spp.	48
	UFJ-LPPV-84	Cerrado	Chapada dos Guimarães, MT	*Echinococcus* spp.	21
	UFJ-LPPV-298	Cerrado	Cuiabá, MT	*Toxocara cati*	8
				*Toxocara canis*	6
*Puma concolor*	UFJ-LPPV-153	Pantanal	Poconé, MT	*Oncicola* spp.	9
	UFJ-LPPV-74	Cerrado	Mineiros, GO	*Toxocara cati*	1
				*Ancylostoma braziliense*	1
	UFJ-LPPV-215	Cerrado	Jataí, GO	*Toxocara cati*	14
	UFJ-LPPV-76	Cerrado	Chapada dos Guimarães, MT	*Ancylostoma pluridentatum*	363
				*Ancylostoma caninum*	21
	UFJ-LPPV-299	Cerrado	Jataí, GO	*Taenia* spp.	14
				*Spirometra* spp.	1
				*Toxocara cati*	2

## Data Availability

Data will be made available on request.

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
