# Peer review of "Gastrointestinal Helminths in Wild Felids in the Cerrado and Pantanal: Zoonotic Bioindicators in Important Brazilian Biomes"

_animals, 2024, doi:10.3390/ani14111622_

Round 1
Reviewer 1 Report
Comments and Suggestions for Authors
The authors have provided and interesting and extensive study of the diversity of helminth parasites present in wild felines from Brazil. I have made specific comments in the different sections of the manuscript which is attached. Major comments include being very clear about how the environment, animals and humans are connected (One Health) and how this interaction affect the transmission of pathogens with zoonotic potential from animals to humans. In some sections of the manuscript, it can be misinterpreted that the authors mean there is a risk for wild animals to be infected with human pathogens. Also, in the discussion, the authors talk about the risk of domestic animals being infected with pathogens from wild animals and vice versa, I would suggest including some of this in the introduction. In general, the study is well conducted and well written, I hope may comments and suggestions help improve the manuscript so that it can be accepted after minor changes.

Author Response
Dear reviewer, we are grateful that you provided your valuable time to review our manuscript. All comments were considered and we hope that we have addressed, we rewrote some parts, created new graphic elements, increased and deleted parts of the original text as can be seen in the new version, marked in yellow. We are available to discuss new aspects, if necessary.
Kind regards.
Reviewer 2 Report
Comments and Suggestions for Authors
In this manuscript, de Sá Moraes et al. examined 27 wild felids that were collected from the Cerrado and Pantanal regions of Brazil and reported the presence of certain gastrointestinal helminths within those carcasses. The taxonomic keys used in this study were properly illustrated in the manuscript and the results were thoroughly analyzed. Nonetheless, there are a few issues that need to be addressed by the authors, which are specified as below:
· The authors should include one additional paragraph in the Introduction section, which briefly describes the classification of parasitic worms and the most distinctive features of each group. Additionally, the different ecosystems represented by the Pantanal and Cerrado regions should be mentioned here, instead of in the Discussion.
· I suggest that the authors slightly rearrange the order of information in the Results section. In my opinion, it is more logical to begin with the morphological insights of different helminths, so that the readers will understand how those parasites were classified based on the structural observations. Then the authors can exhibit the parasite species, abundance and coinfection circumstance within each felid group. The authors do not have to follow exactly my advice, but please make it more comprehensible to the readers.
· The authors solely rely on morphological features to classify those parasitic worms detected from the carcasses. I understand that the taxonomic keys are effective to classify the worms into large categories, such as trematodes, nematodes and cestodes. However, when it comes to smaller divisions, such as genus and species, the classification may become increasingly challenging. To make the current classification result more convincing, the authors may consider performing some analyses at molecular level. For instance, they can extract DNA from the worm samples, amplify some conserved regions, e.g. 18s rDNA, compare the sequences of different samples and build a phylogenetic tree. This result will reflect how accurate the morphology-based classification is.
· The Discussion section of this manuscript should be significantly simplified. Some literature review paragraphs that are irrelevant to the analysis of results, such as Lines 300-325, should be either moved forward to the Introduction section, or completely removed. The authors also used too many paragraphs to analyze each species of parasitic worms they detected from the felids. The arrangement is extremely chaotic. I advise them to paraphrase those discussions, only extract the key conclusions of each cited literature, instead of trying to mention every detail.
· The authors made too much extensions in their Discussion. Some of the contents were out of the scope of this study and cannot be fully supported by the data. For example, the authors kept mentioning how human activities, such as urbanization, deforestation and agricultural expansion can impact the behaviors and survival of wild felids and in turn, the spread of zoonotic diseases. Those points can be briefly mentioned but should not be discussed as exhaustively as in this manuscript. As a matter of fact, the researchers did not include felid samples in this study that were collected from habitats where people rarely visited. Therefore, without such a control group, they cannot attribute the presence of certain parasites in wild felids to human activities or environmental changes. In other words, they should avoid such “over-interpretation” of their data.
Author Response
Dear reviewer, we are grateful that you provided your valuable time to review our manuscript. All comments were considered and we hope that we have addressed all suggestions as follows:
Reviewer: In this manuscript, de Sá Moraes et al. examined 27 wild felids that were collected from the Cerrado and Pantanal regions of Brazil and reported the presence of certain gastrointestinal helminths within those carcasses. The taxonomic keys used in this study were properly illustrated in the manuscript and the results were thoroughly analyzed. Nonetheless, there are a few issues that need to be addressed by the authors, which are specified as below:
Response: Thank you very much.
Reviewer: The authors should include one additional paragraph in the Introduction section, which briefly describes the classification of parasitic worms and the most distinctive features of each group. Additionally, the different ecosystems represented by the Pantanal and Cerrado regions should be mentioned here, instead of in the Discussion.
Response: We have added data in the introduction as seen in yellow and more specifically about biomes in lines 49-57, as well as figures and data in materials and methods. Due to the comments of other reviewers to simplify some parts, we chose not to elaborate on the classification of parasites in the introduction, however we added several aspects of the description of helminths in the results, including more details of morphology and we also increased the discussion by better addressing some groups (markings in yellow in results and discussion).
Reviewer: I suggest that the authors slightly rearrange the order of information in the Results section. In my opinion, it is more logical to begin with the morphological insights of different helminths, so that the readers will understand how those parasites were classified based on the structural observations. Then the authors can exhibit the parasite species, abundance and coinfection circumstance within each felid group. The authors do not have to follow exactly my advice, but please make it more comprehensible to the readers.
Response: We tried to rearrange the text of the results by including the morphological aspects, however we got lost in the three attempts we made, as the primary objective of the study is the study of the parasite community, and the tables and figures were scattered and illogical when allocated later. Moving forward, we created a Venn diagram showing co-infections and restructured the text in this regard as requested.
Reviewer: The authors solely rely on morphological features to classify those parasitic worms detected from the carcasses. I understand that the taxonomic keys are effective to classify the worms into large categories, such as trematodes, nematodes and cestodes. However, when it comes to smaller divisions, such as genus and species, the classification may become increasingly challenging. To make the current classification result more convincing, the authors may consider performing some analyses at molecular level. For instance, they can extract DNA from the worm samples, amplify some conserved regions, e.g. 18s rDNA, compare the sequences of different samples and build a phylogenetic tree. This result will reflect how accurate the morphology-based classification is.
Response: We vehemently agree with what was stated by the reviewer. In this sense, we justify that we identified, up to the species level, only the species in which it was possible to reach that level with morphological attributes and we increased the text and images with detailed data for this purpose, as can be seen in the results section. In species for which identification was not possible, we maintained the classification at the genus level and placed the spp. Our intention is that after publication, we can raise more resources, due to the credit given by the published manuscript, to continue studies on the groups that were identified at the genus level, and thus detail which species they are, carry out the construction of phylogenetic trees and detail the morphological descriptions of these groups in subsequent studies.
Reviewer: The Discussion section of this manuscript should be significantly simplified. Some literature review paragraphs that are irrelevant to the analysis of results, such as Lines 300-325, should be either moved forward to the Introduction section, or completely removed. The authors also used too many paragraphs to analyze each species of parasitic worms they detected from the felids. The arrangement is extremely chaotic. I advise them to paraphrase those discussions, only extract the key conclusions of each cited literature, instead of trying to mention every detail.
Response: At this point, we are confused, as the point contrasts with the position of another reviewer who asked us to increase the discussion with other aspects. We tried to improve some parts, as can be seen in yellow in the text, but if there is a need to reconstruct any other part, we ask that the reviewer feels free to point it out to us and help us bring the best version possible.
Reviewer: The authors made too much extensions in their Discussion. Some of the contents were out of the scope of this study and cannot be fully supported by the data. For example, the authors kept mentioning how human activities, such as urbanization, deforestation and agricultural expansion can impact the behaviors and survival of wild felids and in turn, the spread of zoonotic diseases. Those points can be briefly mentioned but should not be discussed as exhaustively as in this manuscript. As a matter of fact, the researchers did not include felid samples in this study that were collected from habitats where people rarely visited. Therefore, without such a control group, they cannot attribute the presence of certain parasites in wild felids to human activities or environmental changes. In other words, they should avoid such “over-interpretation” of their data.
Response: We rewrote and better restructured the text to explain the point of view presented as can be seen in lines 497-526. New references have also been added to detail the discussed alignment.